# Quantum Oscillator at Temperature *T* and the Evolution of a Charged-Particle State in the Electric Field in the Probability Representation of Quantum Mechanics

**DOI:** 10.3390/e25020213

**Published:** 2023-01-22

**Authors:** Margarita A. Man’ko, Vladimir I. Man’ko

**Affiliations:** 1Lebedev Physical Institute, Leninskii Prospect 53, Moscow 119991, Russia; 2Moscow Institute of Physics and Technology, Dolgoprudnyi, Moscow 141700, Russia; 3Russian Quantum Center, Skolkovo, Moscow 143025, Russia

**Keywords:** tomographic probability distribution, integrals of motion, coherent state, normal probability distribution, entropy

## Abstract

A short review constructing the probability representation of quantum mechanics is given, and examples of the probability distributions describing the states of quantum oscillator at temperature *T* and the evolution of quantum states of a charged particle moving in the electric field of an electrical capacitor are considered. Explicit forms of time-dependent integrals of motion, linear in the position and momentum, are used to obtain varying probability distributions describing the evolving states of the charged particle. Entropies corresponding to the probability distributions of initial coherent states of the charged particle are discussed. The relation of the Feynman path integral to the probability representation of quantum mechanics is established.

## 1. Introduction

In classical mechanics, the states of particles are described by the position *q* and momentum *p*, and in classical statistical mechanics, by the probability density f(q,p) [1,2,3]. In quantum mechanics, the states of particles are described either by complex wave functions ψ(q) [4,5] or by the density matrices [6,7,8]. Other representations of the states, such as Wigner quasidistribution functions W(q,p) [9] and Husimi quasidistribution [10], as well as Glauber–Sudarshan functions and their generalizations, *s*-quasidistributions, are used [11,12]. These representations were found because the attempts to establish the classical description of quantum states, using the probability distributions of the position *q* and momentum *p*, had difficulties associated with the existence of uncertainty relations for the position and momentum.

The evolution of the wave functions of quantum states is described by the Schrödinger equation and the evolution of the density matrices of quantum states is described by the von Neumann equation [13]. The evolution formalism can be also based on the approach of the path integral [14].

Recently, the probability representation of quantum states was introduced [15,16]; see [17], where the states are described by the conventional probability distributions, which are connected to other mentioned representations by integral transforms [18]. To avoid the mentioned difficulties with the Heisenberg uncertainty relations, the probability distribution employed in the approach [15] depends on only one position. In this representation, all quantum effects can be explained, in view of standard properties of conventional probability theory presented in [1].

The evolutions for probability distributions were discussed in [19], and some new relations between the probability distributions connected with calculations of the Franck–Condon principle were obtained in [20].

The aim of this work is to consider different tomographic probability distributions describing the quantum oscillator states and states of a charged particle moving in the electric field *E*, if the initial state at time t=0 is taken to be one of known normalized states, such as the coherent state [11] or the squeezed state, as well as the energy state of the harmonic oscillator or the quantum oscillator at temperature *T*. We obtain explicit expressions for the integrals of motion of a charged particle following [21,22,23]. Using these integrals of motion, we obtain probability distributions describing different states of the charged particle, if the initial state of the particle is prepared in a specific state of a harmonic oscillator, such as the coherent state or excited state.

This paper is organized as follows.

In Section 2, the approach of symplectic tomography is briefly reviewed. In Section 3, tomograms of the oscillator states at temperature *T* are considered. In Section 4, the tomograms of the charged particle with different initial states and path integrals are discussed. A generic approach to the evolution of these states described by explicit forms of tomographic probability distributions is studied in Section 5. The relation of the approach to the Green function of the Schrödinger equation and the path integral is considered in Section 6. In Section 7, the evolution of coherent and excited states of oscillator is studied. Entropies of evolving tomographic probability distributions are discussed in Section 8. Conclusions and prospectives are presented in Section 9.

## 2. Symplectic Tomography

First, we consider tomographic probability distribution describing the particle states in classical statistical mechanics. The conventional description of such states is associated with probability distributions $f\left(q,p,t\right)\geq 0$ in the phase space, i.e., non-negative functions depending (or not depending) on time *t* and on the position *q* and momentum *p* of the particle. This function satisfies the normalization condition
(1)$$\int f\left(q,p,t\right)\;dq\;dp=1.$$One can introduce the classical tomographic probability distribution [24], using the Radon transform [25],
(2)$$w_{\mathrm{cl}}\left(X|\mu ,\nu ,t\right)=\int f\left(q,p,t\right)\;\delta \left(X-\mu q-\nu p\right)\;dq\;dp.$$The inverse Radon transform reads
(3)$$f\left(q,p,t\right)=\frac{1}{4\;\pi^{2}}\;\int w_{\mathrm{cl}}\left(X|\mu ,\nu ,t\right)\;\exp\left[i\left(X-\mu q-\nu p\right)\right]\;dX\;d\mu \;d\nu .$$One has the normalization condition for the non-negative function $w_{\mathrm{cl}}\left(X|\mu ,\nu ,t\right)$; it is
(4)$$\int w_{\mathrm{cl}}\left(X|\mu ,\nu ,t\right)\;dX=1.$$In view of the delta-function properties, the parameter X=μq+νp is the particle position in the reference frames, with rotated and rescaled momentum and position axes in the phase space.

In the case of the quantum particle, we discuss the construction of tomographic probability distribution describing the state of any system, such as an oscillator, with the wave function, where the Hamiltonian reads
(5)$$\hat{H}=\frac{{\hat{p}}^{2}}{2}+V\left(\hat{q},t\right);$$
here, we assume that the particle mass *m* and Planck’s constant *ℏ* are such that m=ℏ=1, and the oscillator frequency is ω=1.

The Schrödinger evolution equation has the form
(6)$$i\;\frac{\partial \psi \left(x,t\right)}{\partial t}=\hat{H}\psi \left(x,t\right),$$
and its solution, in the case of $\psi \left(x,t=0\right)=\psi \left(x\right)$, can be expressed in terms of the Green function $G\left(y,x,t\right)$ as
(7)$$\psi \left(y,t\right)=\int G\left(y,x,t\right)\psi \left(x\right)\;dx.$$

We consider the construction of tomographic conditional probability distribution $w\left(X|\mu ,\nu ,t\right)$ of only one random position *X*; this position is determined as X=μq+νp in the reference frame in the phase space, where the position axis *q* and the momentum axis *p* are transformed according first to rescaling $q^{\prime }=sq$ and $p^{\prime }=s^{-1}p$ and then to the following rotation: $q^{\prime \prime }=\cos\theta q^{\prime }+\sin\theta p^{\prime }$ and $p^{\prime \prime }=-\sin\theta q^{\prime }+\cos\theta p^{\prime }$. This means that the parameter *X* is the position of the charged particle measured in the reference frame in the phase space obtained via rescaling and rotation of the initial axes *q* and *p* determined by real parameters μ and ν. Namely introducing the probability distribution of only one random variable *X* provides the possibility of avoiding the problems connected with the Heisenberg uncertainty relation.

One can easily check that the density operator ${\hat{\rho }}_{\psi }\left(t\right)=|\psi \rangle \langle \psi |$ can be mapped onto the function $w\left(X|\mu ,\nu ,t\right)$ using the operator $\delta \left(X\hat{1}-\mu \hat{q}-\nu \hat{p}\right)$; the map is
(8)$$w\left(X|\mu ,\nu ,t\right)=\mathrm{Tr}\left[{\hat{\rho }}_{\psi }\left(t\right)\delta \left(X\hat{1}-\mu \hat{q}-\nu \hat{p}\right)\right].$$The inverse transform reads [15,17,26,27]
(9)$${\hat{\rho }}_{\psi }\left(t\right)=\frac{1}{2\pi }\int w\left(X|\mu ,\nu ,t\right)\exp\left[i\left(X\hat{1}-\mu \hat{q}-\nu \hat{p}\right)\right].$$In terms of an explicit form of the wave function, relation (Equation 8) converts into
(10)$$w\left(X|\mu ,\nu ,t\right)=\frac{1}{2\pi |\nu |}{\left|\int \psi \left(y,t\right)\exp\left(\frac{i\mu }{2\nu }y^{2}-\frac{iX}{\nu }y\right)\;dy\right|}^{2};$$
this function is non-negative and, in view of (Equation 8), it is obvious that
(11)$$\int w\left(X|\mu ,\nu ,t\right)\;dX=1.$$These properties mean that the function $w\left(X|\mu ,\nu ,t\right)$, called the symplectic tomogram, is the conditional probability distribution; the conditions are labeled by parameters μ and ν, determining the reference frame where the random position *X* is measured. In view of relations (Equation 8) and (Equation 9), the distribution $w\left(X|\mu ,\nu ,t\right)$ (or the state tomogram) contains all information on the density operator (Equation 9). This means that the tomographic probability distribution completely describes the quantum state. Additionally, this is an explicit construction of the probability representation of quantum state ${\hat{\rho }}_{\psi }$, which was not known in the literature.

We can find the Wigner function of the state [9]
(12)$$W_{\psi }\left(q,p,t\right)=\int \psi \left(q+y/2,t\right)\psi^{*}\left(q-y/2,t\right)e^{-ipy}\;dy$$
and check that
(13)$$W_{\psi }\left(q,p,t\right)=\frac{1}{2\pi }\int w\left(X|\mu ,\nu ,t\right)e^{i\left(X-\mu q-\nu p\right)}\;dX\;d\mu \;d\nu .$$The case of parameter s=1 corresponds to the optical tomogram $w\left(X|\theta \right)$, which replaces the symplectic tomogram in (Equation 8) and (Equation 9). Namely, the optical tomogram $w\left(X|\theta \right)$ of photon states is measured in the experiments in quantum optics [28], and it is used to reconstruct the Wigner function [29,30,31].

As became clear, the optical tomogram can also be employed to describe quantum states, and it is not only an experimental tool to measure the object determining the Wigner function, as it was considered before symplectic tomography was introduced. Comparing Equations (Equation 7) and (Equation 9), we obtain a new relation for tomogram, which can be expressed in terms of the Green function of the Schrödinger equation
(14)$$w\left(X|\mu ,\nu ,t\right)=\frac{1}{2\pi |\nu |}{\left|\int G\left(y,x,t\right)\psi \left(x,0\right)\exp\left(\frac{i\mu }{2\nu }y^{2}-\frac{iX}{\nu }y\right)\;dx\;dy\right|}^{2};$$
this formula provides the possibility of mapping the initial wave function $\psi \left(x,0\right)$ onto the evolving tomogram $w\left(X|\mu ,\nu ,t\right)$. It can be rewritten as follows:(15)$$w\left(X|\mu ,\nu ,t\right)={\left|\int g\left(x,\mu ,\nu ,t\right)\psi \left(x,0\right)\;dx\right|}^{2},$$
where
(16)$$g\left(x,\mu ,\nu ,t\right)=\frac{1}{\sqrt{2\pi |\nu |}}\int \exp\left(\frac{i\mu }{2\nu }y^{2}-\frac{iX}{\nu }y\right)G\left(y,x,t\right)\;dy.$$This relation can be used to connect the Feynman path integral approach, which describes the Green function while considering quantum dynamics, with dynamics of the probability distributions determining the system quantum states. To achieve this, one can rewrite the Green function in Equation (Equation 16) in the form of the Feynman path integral, which expresses the Green function in terms of the Feynman path integral [32].

One can associate the probability distribution $w\left(X,\mu ,\nu \right)$ of three random variables with the tomogram, employing the Bayes formula
(17)$$w\left(X,\mu ,\nu \right)=w\left(X|\mu ,\nu \right)P\left(\mu ,\nu \right),$$
where the function $P\left(\mu ,\nu \right)$ is a probability density, e.g., in the form of normal distribution [33],
(18)$$P\left(\mu ,\nu \right)=\frac{1}{\pi }\;e^{-\mu^{2}-\nu^{2}}.$$Analogously, one can also obtain new properties of Hermite polynomials [34].

The density operator $\hat{\rho }$ of the quantum state with such a function is obtained, in view of the formula
(19)$$\hat{\rho }=\frac{1}{2}\int w\left(X,\mu ,\nu \right)e^{\mu^{2}+\nu^{2}}\exp\left[i\left(X-\mu \hat{q}-\nu \hat{p}\right)\right]\;dX\;d\mu \;d\nu .$$The probability distribution $w\left(X,\mu ,\nu \right)$ is determined by the density operator $\hat{\rho }$, using the relation
(20)$$w\left(X,\mu ,\nu \right)=\pi^{-1}\mathrm{Tr}\left[\hat{\rho }\;e^{-\mu^{2}-\nu^{2}}\delta \left(X-\mu \hat{q}-\nu \hat{p}\right)\right];$$
the function $w\left(X,\mu ,\nu \right)$ is non-negative, and it satisfies the normalization condition
(21)$$\int w\left(X,\mu ,\nu \right)\;dX\;d\mu \;d\nu =1.$$Thus, the quantum state is described by the probability distribution of three random variables: *X*, μ, and ν, with *X*, being the position, and parameters μ and ν being random characteristics of reference frames in the particle phase space, where the position *X* is measured. We can use different functions $P\left(\mu ,\nu \right)$ in (Equation 17) and (Equation 18).

Born’s rule for the probability $P_{2}^{1}=\mathrm{Tr}\left({\hat{\rho }}_{1}{\hat{\rho }}_{2}\right)$ reads
(22)$$P_{2}^{1}=\pi^{-2}\int w_{1}\left(X,\mu ,\nu \right)w_{2}\left(Y,-\mu ,-\nu \right)\exp\left[i\left(X+Y\right)+2\mu^{2}+2\nu^{2}\right]\;dX\;dY\;d\mu \;d\nu .$$If the probability distribution $P\left(\mu ,\nu \right)$ is chosen to be
(23)$$P\left(\mu ,\nu \right)\equiv P_{0}\left(\mu ,\nu \right)=\delta \left(\mu -\mu_{0}\right)\delta \left(\nu -\nu_{0}\right),$$
the probability distribution $w\left(X,\mu ,\nu \right)$ takes the form
(24)$$w_{0}\left(X,\mu ,\nu \right)=w\left(X|\mu ,\nu \right)\delta \left(\mu -\mu_{0}\right)\delta \left(\nu -\nu_{0}\right);$$
this means that the probability distribution of one random position reads
(25)$$\int w_{0}\left(X,\mu ,\nu \right)\;d\mu \;d\nu =w\left(X|\mu_{0},\nu_{0}\right),$$
and it coincides with the tomographic conditional probability distribution describing the quantum state with the density operator $\hat{\rho }\left(t=0\right)$ (Equation 8).

If the probability distribution $P\left(\mu ,\nu \right)$ is chosen as the function of two random variables μ and ν, the probability distribution of the position *X* reads
(26)$$w_{P}\left(X\right)=\int w_{0}\left(X|\mu ,\nu \right)P\left(\mu ,\nu \right)\;d\mu \;d\nu .$$

## 3. Tomograms of the Oscillator States at Temperature *T*

Now, we consider the tomogram of the oscillator state, corresponding to the thermal state with temperature *T*, introducing the parameter β=1/T and assuming the oscillator parameters and Planck’s constant to be m=ω=ℏ=1. The density operators of the thermal state with the Hamiltonian $\hat{H}=\frac{{\hat{p}}^{2}}{2}+\frac{{\hat{q}}^{2}}{2}\;,$ i.e., $\hat{\rho }\left(\beta \right)=Z^{-1}\left(\beta \right)\exp\left(-\hat{H}\beta \right)$ has, in the position representation, the matrix elements of the Gaussian form $\langle x|\hat{\rho }\left(\beta \right)|x^{\prime }\rangle =\rho \left(x,x^{\prime },\beta \right)$ [32]; they are
(27)$$\rho \left(x,x^{\prime },\beta \right)=\frac{1}{Z(\beta )}\sqrt{\frac{1}{2\;\pi \sinh\beta }}\;\exp\left\{-\frac{1}{2\;\sinh\beta }\left[\left(x^{2}+x^{\prime 2}\right)\cosh\beta -2\;xx^{\prime }\right]\right\}.$$The density operator $\hat{\rho }\left(\beta \right)$ satisfies the condition $\mathrm{Tr}\left[\hat{\rho }\left(\beta \right)\right]=1$; this means that, for the oscillator at temperature $\beta^{-1}=T$, we have
(28)$$Z\left(\beta \right)=\frac{e^{-1/2T}}{1-e^{-1/T}}=\frac{1}{2\;\sinh(1/2T)}.$$In the Gaussian state (Equation 27), the mean values of the position $\langle q\rangle =\mathrm{Tr}\left(\hat{q}\hat{\rho }\left(\beta \right)\right)$ and the momentum $\langle p\rangle =\mathrm{Tr}\left(\hat{p}\hat{\rho }\left(\beta \right)\right)$ are equal to zero. Additionally, the dispersion of the position ${\left(\delta q\right)}^{2}=\mathrm{Tr}\left({\hat{q}}^{2}\hat{\rho }\left(\beta \right)\right)$ and the dispersion of the momentum ${\left(\delta p\right)}^{2}=\mathrm{Tr}\left({\hat{p}}^{2}\hat{\rho }\left(\beta \right)\right)$ read
(29)$${\left(\delta q\right)}^{2}=\frac{1}{2}\;\coth\left(\beta /2\right),\;{\left(\delta p\right)}^{2}=\frac{1}{2}\;\coth\left(\beta /2\right).$$In addition, the correlation parameter *r* of the position and the momentum is also equal to zero, due to the equality
(30)$$\mathrm{Tr}\left[\left(\hat{q}\hat{p}+\hat{p}\hat{q}\right)\hat{\rho }\left(\beta \right)\right]=0.$$Thus, the Schrödinger–Robertson uncertainty relation of the position and momentum [35,36]
(31)$${\left(\delta q\right)}^{2}{\left(\delta p\right)}^{2}\geq \frac{1}{4}\frac{1}{1-r^{2}}$$
corresponds to the inequality for the state (Equation 27), which reads
(32)$$\frac{1}{4}{\left[\coth\left(\beta /2\right)\right]}^{2}\geq \frac{1}{4}\;.$$Tomogram of the thermal state
(33)$$w\left(X|\mu ,\nu ,\beta \right)=\mathrm{Tr}\left[\hat{\rho }\left(\beta \right)\delta \left(X-\mu \hat{q}-\nu \hat{p}\right)\right]$$
has the form of conditional normal probability distribution
(34)$$\begin{array}{c} \bar{X}=\mu \langle q\rangle +\nu \langle p\rangle =0, \end{array}$$
(35)$$\begin{array}{c} \sigma =\frac{1}{2}\left(\mu^{2}+\nu^{2}\right)\coth\left(\beta /2\right). \end{array}$$Then, the symplectic tomogram identified with the oscillator state at temperature *T* is explicitly described as follows:(36)$$w\left(X|\mu ,\nu ,T\right)=\frac{1}{\sqrt{\pi \left(\mu^{2}+\nu^{2}\right)\coth\left(1/2T\right)}}\exp\left(-\frac{X^{2}}{\left(\mu^{2}+\nu^{2}\right)\coth\left(1/2T\right)}\right).$$For zero temperature, T=0, the parameter $\coth\left(\beta /2\right)=1$, and the state with density operator (Equation 27) is the ground state of the oscillator with the tomogram
(37)$$w_{0}\left(X|\mu ,\nu ,T=0\right)=\frac{1}{\sqrt{\pi (\mu^{2}+\nu^{2}}}\exp\left(-\frac{X^{2}}{\mu^{2}+\nu^{2}}\right).$$The oscillator tomographic probability distribution of three random variables *X*, μ, and ν reads
(38)$$w\left(X,\mu ,\nu ,T\right)=\frac{\sqrt{\pi }}{\sqrt{\left(\mu^{2}+\nu^{2}\right)\coth\left(1/2T\right)}}\exp\left[-\frac{X^{2}}{\left(\mu^{2}+\nu^{2}\right)\coth\left(1/2T\right)}-\left(\mu^{2}+\nu^{2}\right)\right];$$
this probability distribution describes the state of the quantum oscillator at temperature *T*.

For generic probability distribution $P\left(\mu ,\nu \right)$, we have the tomogram
(39)$$w_{P}\left(X,\mu ,\nu ,T\right)=\frac{1}{\sqrt{\left(\mu^{2}+\nu^{2}\right)\coth\left(1/2T\right)}}\exp\left[-\frac{X^{2}}{\left(\mu^{2}+\nu^{2}\right)\coth\left(1/2T\right)}\right]P\left(\mu ,\nu \right).$$

Tomogram of the ground state of harmonic oscillator at temperature *T*, with the wave function $\psi_{0}\left(x\right)=\pi^{-1/4}\exp\left(-x^{2}/2\right)$, given by the integral
(40)$$w_{0}\left(X|\mu ,\nu \right)=\frac{1}{2\pi |\nu |}{\left|\int \pi^{-1/4}\exp\left(-\frac{y^{2}}{2}+\frac{i\mu }{2\nu }y^{2}-\frac{iXy}{\nu }\right)dy\right|}^{2},$$
has the form
(41)$$w_{0}\left(X|\mu ,\nu \right)=\frac{1}{\sqrt{\pi (\mu^{2}+\nu^{2})}}\exp\left(-\frac{X^{2}}{\mu^{2}+\nu^{2}}\right).$$For the function $P\left(\mu ,\nu \right)=\pi^{-1}\exp\left(-\mu^{2}-\nu^{2}\right)$, the probability distribution of three random variables *X*, μ, and ν reads
(42)$$w_{0}\left(X,\mu ,\nu \right)=\frac{\pi^{-3/2}}{\sqrt{\mu^{2}+\nu^{2}}}\exp\left(-\frac{X^{2}}{\mu^{2}+\nu^{2}}-\mu^{2}-\nu^{2}\right);$$
this means that the probability distribution of position *X* has the form
(43)$$w_{0}\left(X\right)=\pi^{-3/2}\int \frac{1}{\sqrt{\mu^{2}+\nu^{2}}}\exp\left(-\frac{X^{2}}{\mu^{2}+\nu^{2}}-\mu^{2}-\nu^{2}\right)\;d\mu \;d\nu .$$

In the case of the classical oscillator at temperature *T*, with the joint probability density
(44)$$f\left(q,p\right)=\frac{1}{Z(T)}e^{(-p^{2}-q^{2})/2T},\;Z\left(T\right)=\frac{2\;\pi }{T}\;,$$
the symplectic tomogram is
(45)$$w_{\mathrm{cl}}\left(X|\mu ,\nu ,T\right)=\int f\left(q,p\right)\;\delta \left(X-\mu q-\nu p\right)\;dq\;dp;$$
in the explicit form, we have
(46)$$w_{\mathrm{cl}}\left(X|\mu ,\nu ,T\right)=\frac{1}{\sqrt{2\;\pi (\mu^{2}+\nu^{2})T}}\exp\left[-\frac{X^{2}}{2\left(\mu^{2}+\nu^{2}\right)T}\right].$$Employing the relationship
(47)$$\lim_{\varepsilon \to 0}\frac{1}{\sqrt{\pi \varepsilon }}\;e^{-x^{2}/\varepsilon }=\delta \left(x\right),\;\varepsilon =2T,$$
in the limit T→0, we arrive at the probability density, which is
(48)$$w_{\mathrm{cl}}\left(q,p,T=0\right)=\delta \left(q\right)\delta \left(p\right).$$This means that for the classical state, ${\left(\delta q\right)}^{2}{\left(\delta p\right)}^{2}=0$, that is permitted for the classical oscillator, since the uncertainty relations for the position and momentum take place for the quantum oscillator and not for the classical oscillator.

The classical tomographic probability distribution of three random variables *X*, μ, and ν, at temperature *T*, reads
(49)$$w_{\mathrm{cl}}\left(X,\mu ,\nu ,T\right)=\frac{\sqrt{\pi }}{\sqrt{2\;\left(\mu^{2}+\nu^{2}\right)T}}\exp\left[-\frac{X^{2}}{2\left(\mu^{2}+\nu^{2}\right)T}-\left(\mu^{2}+\nu^{2}\right)\right].$$

## 4. Charge in the Electric Field

The integrals of motion for classical systems such as, for example, a charged particle with the mass m=1 and charge *e* moving in the constant electric field *E* along the field are obtained, in view of formulas for the position q(t) and momentum p(t) of the charge, which read
(50)$$\begin{array}{ccc} q(t) & = & q_{0}+p_{0}t+at^{2}/2, \end{array}$$
(51)$$\begin{array}{ccc} p(t) & = & p_{0}+at, \end{array}$$
where a=eE is the charge acceleration. The initial position $q_{0}$ in (Equation 50) and momentum $p_{0}$ in (Equation 51) of the charge trajectory in its phase space are the constants of motion, which are functions of time and the position *q* and momentum *p* of the form
(52)$$\begin{array}{ccc} q_{0}\left(t\right) & = & q-pt+at^{2}/2, \end{array}$$
(53)$$\begin{array}{ccc} p_{0}\left(t\right) & = & p-at. \end{array}$$Thus, we can check that for the classical charge *e*, the integrals of motion $q_{0}\left(t\right)$ and $p_{0}\left(t\right)$, corresponding to the classical Hamiltonian $H=\frac{p^{2}}{2}-eEq$, are linear forms of the position *q*, momentum *p*, and time *t*.

For the charged particle with the charge *e* and Hamiltonian (Equation 5), where $V\left(\hat{q},t\right)=-a\hat{q}$ and the constant parameter a=eE, there exist two integrals of motion ${\hat{q}}_{0}\left(t\right)=\hat{q}-\hat{p}t+at^{2}/2$ and ${\hat{p}}_{0}\left(t\right)=\hat{p}-at.$

Since the Green function $G\left(y,x,t\right)$ [22,23] satisfies the system of equations
(54)$$\begin{array}{c} \left(y+it\;\frac{\partial }{\partial y}+\frac{at^{2}}{2}\right)G\left(y,x,t\right)=x\;G\left(y,x,t\right), \end{array}$$
(55)$$\begin{array}{c} \left(-i\;\frac{\partial }{\partial y}-at\right)G\left(y,x,t\right)=i\;\frac{\partial }{\partial x}G\left(y,x,t\right), \end{array}$$
(56)$$\begin{array}{c} i\;\frac{\partial G\left(y,x,t\right)}{\partial t}=\left(-\frac{1}{2}\frac{\partial^{2}}{\partial y^{2}}-a\;y\right)G\left(y,x,t\right), \end{array}$$
with the initial conditions $G\left(y,x,t=0\right)=\delta \left(y-x\right)$, it can be found in the Gaussian form. This function can be presented in the limit of integral form
(57)$$G\left(y,x,t\right)=\lim_{n\to \infty }\left[\int_{u_{1}}\int_{u_{2}}\cdots \int_{u_{n}}u_{y,x_{n}}\left(t_{n}\right)u_{x_{n},x_{n-1}}\left(t_{n-1}\right)\cdots u_{x_{1},x_{0}}\left(t_{1}\right)\prod_{k=1}^{n}\;dx_{k}\right],$$
where $u_{x_{n},x_{n-1}}\left(t_{n}\right)$ is the matrix element of the evolution operator, which is the Green function $G\left(x_{n},x_{n-1},t_{n},t_{n-1}\right)$.

One can check that the Gaussian solution for the Green function satisfying (Equation 54)–(Equation 56) reads
(58)$$G\left(y,x,t\right)=f\left(t\right)\exp\left\{i\left[\frac{{\left(y-x-at^{2}/2\right)}^{2}}{2t}+xat\right]\right\},$$
where the function f(t) is determined by (Equation 56).

Following the consideration of quantum integrals of motion for the systems with the Hamiltonians, which are quadratic forms in the position and momentum, presented in [21,22], we can use quantized expressions in (Equation 54)–(Equation 56) with operators ${\hat{q}}_{0}\left(t\right)$, ${\hat{p}}_{0}\left(t\right)$, $\hat{q}$, and $\hat{p}$ acting on the wave functions in the system with the charge Hamiltonian $\hat{H}=\frac{{\hat{p}}^{2}}{2}-a\hat{q}$.

In the case of time-dependent electric field *E*, the parameter *a* is the corresponding function, which depends on the time. If we find the quantum integrals of motion ${\hat{q}}_{0}\left(t\right)$ and ${\hat{p}}_{0}\left(t\right)$, we obtain the system of equations for the Green function of the Schrödinger equation related to the integrals of motion.

## 5. Evolution of the Density Operator of Quantum System
in the Probability Representation

The state vectors ∣ψ(t)〉 of the pure quantum states satisfy the Schrödinger Equation (Equation 6). The density operator $\hat{\rho }\left(t\right)$ of the quantum states satisfies the von Neumann equation for the systems with the Hamiltonian of the form
(59)$$\frac{\partial \hat{\rho }\left(t\right)}{\partial t}+i\left[\hat{H}\left(t\right),\hat{\rho }\left(t\right)\right]=0.$$The density operator $\hat{\rho }\left(t\right)$ can be found as a solution of this equation, if one knows the evolution operator $\hat{u}\left(t\right)$ satisfying the following equation:(60)$$i\;\frac{\partial \hat{u}\left(t\right)}{\partial t}=\hat{H}\left(t\right)\hat{u}\left(t\right);\;\hat{u}\left(t=0\right)=\hat{1}.$$For the known density operator $\hat{\rho }\left(t=0\right)=\hat{\rho }\left(0\right)$, the evolution density operator $\hat{\rho }\left(t\right)$ reads
(61)$$\hat{\rho }\left(t\right)=\hat{u}\left(t\right)\hat{\rho }\left(0\right){\hat{u}}^{†}\left(t\right).$$If the Hamiltonian $\hat{H}\left(t\right)$ is the Hermitian operator, i.e., ${\hat{H}}^{†}\left(t\right)=\hat{H}\left(t\right)$, the evolution operator $\hat{u}\left(t\right)$ is the unitary operator, i.e., ${\hat{u}}^{†}\left(t\right)\hat{u}\left(t\right)=\hat{u}\left(t\right){\hat{u}}^{†}\left(t\right)=\hat{1}$.

If the Hamiltonian $\hat{H}\left(t\right)$ of the system under consideration does not depend on time, the evolution operator $\hat{u}\left(t\right)$ satisfying Equation (Equation 60) reads
(62)$$\hat{u}\left(t\right)=\exp\left(-it\hat{H}\right).$$Any operators $\hat{I}\left(t\right)$ of the form
(63)$$\hat{I}\left(t\right)=\hat{u}\left(t\right)\hat{I}\left(0\right){\hat{u}}^{†}\left(t\right)$$
are integrals of motion, i.e.,
(64)$$\frac{d\hat{I}\left(t\right)}{dt}=\frac{\partial \hat{I}\left(t\right)}{\partial t}+i\left[\hat{H},\hat{I}\left(t\right)\right]=0,$$
and this means that $\langle \hat{I}\left(0\right)\rangle$ satisfy the condition
(65)$$\mathrm{Tr}\left[\hat{\rho }\left(t\right)\hat{I}\left(0\right)\right]=\mathrm{Tr}\left[\hat{\rho }\left(0\right){\hat{I}}_{H}\left(t\right)\right].$$The density operator $\hat{\rho }\left(t\right)$ satisfying the evolution Equation (Equation 59) is the integral of motion. Additionally, the operators
(66)$${\hat{q}}_{0}\left(t\right)=\hat{u}\left(t\right)\hat{q}{\hat{u}}^{†}\left(t\right),\;{\hat{p}}_{0}\left(t\right)=\hat{u}\left(t\right)\hat{p}{\hat{u}}^{†}\left(t\right)$$
where $\hat{q}$ and $\hat{p}$ are the position and momentum operators, are the integrals of motion. Then, the operators
(67)$${\hat{q}}_{H}\left(t\right)={\hat{u}}^{†}\left(t\right)\hat{q}\hat{u}\left(t\right),\;{\hat{p}}_{H}\left(t\right)={\hat{u}}^{†}\left(t\right)\hat{p}\hat{u}\left(t\right)$$
are the position and momentum operators satisfying, in the Heisenberg representation, the following evolution equations:(68)$$\frac{\partial {\hat{q}}_{H}\left(t\right)}{\partial t}-i\left[\hat{H},{\hat{q}}_{H}\left(t\right)\right]=0,\;\frac{\partial {\hat{p}}_{H}\left(t\right)}{\partial t}-i\left[\hat{H},{\hat{p}}_{H}\left(t\right)\right]=0.$$For systems with time-independent Hamiltonians, the integrals of motion ${\hat{I}}_{0}\left(t\right)$ (Equation 64) and the operators ${\hat{I}}_{H}\left(t\right)={\hat{u}}^{†}\left(t\right)I\left(0\right)\hat{u}\left(t\right)$ have the property
(69)$${\hat{I}}_{H}\left(t\right)={\hat{I}}_{0}\left(-t\right);$$
this means that, if one has an explicit form of the integrals of motion ${\hat{I}}_{0}\left(t\right)$, the corresponding operator, in the Heisenberg representation, is also given as the integral of motion for the negative time −t, and vice versa. The discussed properties provide the possibility of finding the evolution of the symplectic tomogram, e.g., of the oscillator, which is determined by the density operator $\hat{\rho }\left(t\right)$; it is
(70)$$w\left(X|\mu ,\nu ,t\right)=\mathrm{Tr}\left[\hat{\rho }\left(t\right)\delta \left[X-\mu \hat{q}-\nu \hat{p}\right)\right].$$In fact, the density operator is the solution of (Equation 59); from this fact follows that, in view of (Equation 65), we arrive at
(71)$$w\left(X|\mu ,\nu ,t\right)=\mathrm{Tr}\left[\hat{u}\left(t\right)\hat{\rho }\left(0\right){\hat{u}}^{†}\left(t\right)\delta \left[X-\mu \hat{q}-\nu \hat{p}\right)\right],$$
or
(72)$$w\left(X|\mu ,\nu ,t\right)=\mathrm{Tr}\left[\hat{\rho }\left(0\right)\delta \left[X-\mu {\hat{q}}_{H}\left(t\right)-\nu {\hat{p}}_{H}\left(t\right)\right)\right].$$

## 6. The State Evolution of the Oscillator and Charge Probability Distributions

Employing expression (Equation 72), we study the time dependence of tomographic probability distributions of two systems–oscillators and their charge in the electric field.

The state evolution can be presented as the time evolution of the tomographic probability distribution determining the quantum state. This evolution can be obtained using the evolution equation for the probability distribution, which follows from the evolution equation for the density operator (Equation 59). In our study, we employ the possibility of using the evolution equation for the case of Hamiltonians, which are quadratic forms of the position and momentum operators. For such systems, the Heisenberg position and momentum are expressed as linear forms in the position and momentum, and we use this circumstance to explicitly obtain the evolution of the tomogram due to Equation (Equation 72).

First, we consider the oscillator initially prepared in the state of thermal equilibrium at temperature *T*, then at time t=0, it is removed from the potential $V\left(\hat{q}\right)={\hat{q}}^{2}/2$ and starts to freely move. The evolution of such a state of the oscillator is described by (Equation 72), where the initial density operator is associated with the tomographic probability distribution (Equation 36). For free motion of quantum oscillator with a removed spring,
(73)$${\hat{q}}_{H}\left(t\right)=\hat{q}+\hat{p}\;t,\;{\hat{p}}_{H}\left(t\right)=\hat{p},$$
and one has the equality
(74)$$X-\mu {\hat{q}}_{H}\left(t\right)-\nu {\hat{p}}_{H}\left(t\right)=X-\mu \left(\hat{q}+\hat{p}\;t\right)-\nu \hat{p}.$$Then, one can express the argument of Dirac delta-function as follows:(75)$$\tilde{X}-\tilde{\mu }\hat{q}-\tilde{\nu }\hat{p}=X-\mu \hat{q}-\left(\nu +\mu t\right)\hat{p}.$$Thus, expression (Equation 72) describes the evolution of tomographic conditional probability distribution of the oscillator state initially given for time t=0 by (Equation 36), providing at time *t* the following result:(76)$$w\left(X|\mu ,\nu ,T,t\right)=\frac{1}{\sqrt{\pi \left[\left(\mu^{2}+{\left(\nu +\mu t\right)}^{2}\right)\coth\left(1/2T\right)\right]}}\exp\left[-\frac{X^{2}}{\left(\mu^{2}+{\left(\nu +\mu t\right)}^{2}\right)\coth\left(1/2T\right)}\right].$$At zero temperature T=0, we have the evolution of the tomogram of the oscillator’s initial ground state, if the potential energy $V\left(\hat{q}\right)={\hat{q}}^{2}/2$ is removed (say, a spring of the oscillator is suddenly removed at time t=0) and the probability distribution describing the evolving state of the oscillator is given by normal time-dependent distribution of the form
(77)$$w_{0}\left(X|\mu ,\nu ,t\right)=\frac{1}{\sqrt{\pi \left(\mu^{2}+{\left(\nu +\mu t\right)}^{2}\right)}}\exp\left[-\frac{X^{2}}{\mu^{2}+{\left(\nu +\mu t\right)}^{2}}\right].$$At the initial large temperatures, we obtain the expression corresponding to the classical behavior of the probability of the oscillator state.

Second, we consider a case in which the initial state of the charged oscillator at temperature *T* and time t=0 starts to evolve following the Hamiltonian with potential energy $V\left(\hat{q}\right)=-a\hat{q}$. Here, equality (Equation 72) converts into
(78)$$X-\mu {\hat{q}}_{H}\left(t\right)-\nu {\hat{p}}_{H}\left(t\right)=\tilde{X}-\tilde{\mu }\hat{q}-\tilde{\nu }\hat{p}=X-\mu \left(\hat{q}+\hat{p}\;t+at^{2}/2\right)-\nu \left(\hat{p}+at\right).$$In view of the relations
(79)$$\tilde{X}=X-\mu \left(at^{2}/2\right)-\nu at,\;\tilde{\mu }=\mu ,\;\tilde{\nu }=\nu +\mu t,$$
after replacing the parameters *X*, μ, and ν in Equation (Equation 36) with the above parameters $\tilde{X}$, $\tilde{\mu }$, and $\tilde{\nu }$, we arrive at the expression of the tomogram of the time-dependent state of the charged oscillator evolving in the potential $V\left(\hat{q}\right)=-eE\hat{q}$. As a result, tomogram $w_{e}$ of normal distribution describing the evolution of thermal state reads
(80)$$w_{e}\left(X|\mu ,\nu ,T\right)=\frac{1}{\sqrt{\pi \left[\left(\mu^{2}+{\left(\nu +\mu t\right)}^{2}\right)\coth\left(1/2T\right)\right]}}\exp\left[-\frac{{\left[X-\mu \left(at^{2}/2\right)-\nu at\right]}^{2}}{\sqrt{\left(\mu^{2}+{\left(\nu +\mu t\right)}^{2}\right)\coth\left(1/2T\right)}}\right].$$Thus, for the charged oscillator state, we obtain a tomogram in the form of normal distribution with an evolving mean value of the quadrature *X* and evolving dispersion of the quadrature.

## 7. Evolution of Coherent and Excited States of Oscillator

Now, we will discuss the evolution of other important states of the charged oscillator. Assume that the oscillator initial state is chosen either as the coherent state with the wave function $\psi_{\alpha }\left(x\right)$, such that $\hat{a}\psi_{\alpha }\left(x\right)=\alpha \psi_{\alpha }\left(x\right)$, where $\hat{a}=2^{-1/2}\left(\hat{q}+i\hat{p}\right)$; or the state $\psi_{n}\left(x\right)$, such that ${\hat{a}}^{†}\hat{a}\psi_{n}\left(x\right)=n\psi_{n}\left(x\right);\;n=1,2,3,\ldots$ Additionally, it is common knowledge that
(81)$$\begin{array}{ccc} \psi_{\alpha }\left(x\right) & = & \pi^{-1/4}\exp\left(-\frac{{\left|\alpha \right|}^{2}}{2}-\frac{x^{2}}{2}+\sqrt{2}\;\alpha x-\frac{\alpha^{2}}{2}\right), \end{array}$$
(82)$$\begin{array}{ccc} \psi_{n}\left(x\right) & = & \frac{\pi^{-1/4}e^{-x^{2}/2}}{\sqrt{2^{n}n!}}\;H_{n}\left(x\right). \end{array}$$In [26], it is shown that, for an arbitrary normalized wave function ψ(x), tomogram $w_{\psi }\left(X|\mu ,\nu \right)$ reads
(83)$$w_{\psi }\left(X|\mu ,\nu \right)=\frac{1}{2\pi |\nu |}{\left|\int \psi \left(y\right)\exp\left(\frac{i\mu }{2\nu }y^{2}-\frac{iX}{\nu }y\right)\;dy\right|}^{2}.$$In view of this expression, we can obtain symplectic tomograms of the oscillator state at time t=0 for (Equation 81) and (Equation 82), as follows:

For the coherent state, it reads
(84)$$w_{\alpha }\left(X|\mu ,\nu \right)=\frac{1}{\sqrt{2\pi \sigma_{\alpha }}}\exp\left(-\frac{{\left[X-\bar{X}\left(\alpha \right)\right]}^{2}}{2\sigma_{\alpha }}\right),$$
where ${\bar{X}}_{\alpha }=\sqrt{2}\;\mu \;Re\;\alpha +\sqrt{2}\;\nu \;Im\;\alpha$ and $\sigma_{\alpha }=\left(\mu^{2}+\nu^{2}\right)/2$.

For the excited state, it is
(85)$$w_{n}\left(X|\mu ,\nu \right)=\frac{1}{\sqrt{\pi \left(\mu^{2}+\nu^{2}\right)}}\exp\left(-\frac{X^{2}}{\mu^{2}+\nu^{2}}\right)\frac{1}{2^{n}n!}{\left|H_{n}\left(\frac{X}{\mu^{2}+\nu^{2}}\right)\right|}^{2}.$$

Thus, employing the tool elaborated for obtaining the symplectic tomogram evolution for the charged oscillator moving in the constant electric field by replacing the parameters *X*, μ, and ν with the parameters $\tilde{X}$, $\tilde{\mu }$, and $\tilde{\nu }$, we can easily derive the symplectic tomogram evolution for the initial states (Equation 84) and (Equation 85).

For the coherent state, it reads
(86)$$w_{\alpha }\left(X|\mu ,\nu ,t\right)=\frac{1}{\sqrt{\pi \left[\mu^{2}+{\left(\nu +\mu t\right)}^{2}\right]}}\exp\left(-\frac{{\left[X-\bar{X}-\mu \left(at^{2}/2\right)-\nu at\right]}^{2}}{\mu^{2}+{\left(\nu +\mu t\right)}^{2}}\right),$$
where $\bar{X}=\sqrt{2}\;\tilde{\mu }\;Re\;\alpha +\sqrt{2}\;\tilde{\nu }\;Im\;\alpha$, $\tilde{\mu }=\mu$, and $\tilde{\nu }=\nu +\mu t$.

For the excited state, it is
(87)$$\begin{array}{ccc} w_{n}\left(X|\mu ,\nu ,t\right) & = & \frac{1}{\sqrt{\pi \left(\mu^{2}+{\left(\nu +\mu t\right)}^{2}\right)}}\exp\left(-\frac{{\left[X-\mu (at^{2}/2)-\nu at\right]}^{2}}{\mu^{2}+{\left(\nu +\mu t\right)}^{2}}\right)\\ & & \times \frac{1}{2^{n}n!}\;{\left|H_{n}\left(\frac{X-\mu (at^{2}/2)-\nu at}{\mu^{2}+{\left(\nu +\mu t\right)}^{2}}\right)\right|}^{2}. \end{array}$$

## 8. Entropies of Evolving Tomographic Probability Distributions

Since the states of systems are identified with conditional probability distributions, we can introduce such characteristics of the states as Shannon entropy, which is related to the symplectic tomogram as follows:(88)$$S\left(\mu ,\nu \right)=-\int w\left(X|\mu ,\nu \right)\;\ln\;w\left(X|\mu ,\nu \right)\;dX.$$For tomogram $w_{P}\left(X,\mu ,\nu \right)=w\left(X|\mu ,\nu \right)P\left(\mu ,\nu \right)$ depending on three random variables, Shannon entropy reads
(89)$$S=-\int w\left(X|\mu ,\nu \right)\;P\left(\mu ,\nu \right)\left[\ln w\left(X|\mu ,\nu \right)+\ln P\left(\mu ,\nu \right)\right]\;dX\;d\mu \;d\nu .$$Such entropies are new characteristics of quantum states of such systems as an oscillator. The entropy has the form of sum $S=S_{1}+S_{2}$, with
(90)$$\begin{array}{ccc} S_{1} & = & -\int w\left(X,\mu ,\nu \right)\;\ln\;w\left(X|\mu ,\nu \right)\;dX\;d\mu \;d\nu , \end{array}$$
(91)$$\begin{array}{ccc} S_{2} & = & -\int w\left(X,\mu ,\nu \right)\;\ln\;P\left(\mu ,\nu \right)\;dX\;d\mu \;d\nu . \end{array}$$

The von Neumann entropy $-\mathrm{Tr}\left(\hat{\rho }\;\ln\;\hat{\rho }\right)$ can be expressed in terms of a symplectic tomogram, in view of the relationship
(92)$$\hat{\rho }=\frac{1}{2\pi }\int w\left(X|\mu ,\nu \right)\;\exp\left[i\left(X-\mu \hat{q}-\nu \hat{p}\right)\right]\;dX\;d\mu \;d\nu .$$

Meanwhile, the developed approach can be used to study the properties of multipartite systems, where the probability distributions of many random variables describe the quantum states. For example, the entanglement phenomenon can be related to the properties of probability distributions; we will consider this aspect in a future publication.

Thus, employing recently introduced probability representation of quantum states, we can clarify different aspects of randomness in quantum mechanics.

## 9. Conclusions

To conclude, we point out the main results of this work.

Throughout the last century, there were many attempts to formulate various notions of quantum states using different concepts of the wave function, density operator, and quasidistribution functions such as the Wigner function; thus, the possibility of describing the notion of quantum state and identifying the state with a conventional probability distribution function was recently realized [17]. In this paper, we present a short review of this approach based on the existence of an invertible map of the density operator; for example, of the oscillator state onto the tomographic conditional probability distribution function. Such a probability distribution also exists for the classical oscillator, but the difference in the probability distributions is connected to the existence of the uncertainty relations of the position and momentum in the quantum case and the violation of the uncertainty relation in the classical case. These aspects are the reason why the construction of the probability representation of quantum mechanics took so many decades.

We point out that an analogous consideration can be extended to the case of systems with discrete variables such as spin systems. The spin states can be described by probability distributions, and these probability distributions can also be investigated [37,38].

The other new important result of this work is introducing entropy of the quantum state, which is conventional Shannon entropy of the introduced probability representation. We also considered quantum statistics and constructed the conventional probability distribution identified with the quantum state at temperature *T*. The explicit form of the harmonic oscillator state identified by the probability distribution is given by Equation (Equation 36). Since this distribution is normal probability distribution, we obtained all its characteristics and the results can be extended to the case of many systems.

Additionally, another important aspect of quantum mechanics is the relation of the Green function of the Schrödinger equation to the Feynmann path integral. We also found the relation of the evolution of the probability distribution of the harmonic oscillator to the path integral (the Feynmann path integral) on the example of a harmonic oscillator.

The different kinds of entropic inequalities known in the probability theory can be discussed for the introduced tomographic probability distributions, and we plan to do this in a future publication. Additionally, the probability representation of quantum mechanics provides the possibility of introducing different distributions and relating them to the irreducible representations of symmetry groups [33] such as unitary SU(n) symmetry of a harmonic oscillator, relating the approach to construction of the group representation operators acting in Hilbert spaces of the wave functions in the oscillator space; this will be considered in the next publication, along with the generalization of the elaborated approach to study the properties of quantum field states.

## Data Availability

Not applicable.

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
