# Peer review of "Quantum Oscillator at Temperature T and the Evolution of a Charged-Particle State in the Electric Field in the Probability Representation of Quantum Mechanics"

_entropy, 2023, doi:10.3390/e25020213_

Round 1

Reviewer 1 Report

This paper continues the long-term research of the authors on the probability representation of quantum mechanics. That is an interesting topic that has gained interest in the quantum optics community over the years.

The paper contains a couple of new results: the notion of entropy in such an approach (paralleling the conventional Shanon definition)  and a nice relation between the evolution of the probability distribution of the harmonic oscillator to the Feynmann path integral.  All in all, the results are beautiful and I recommend publication as it is of this paper.

Author Response

We are grateful to the first referee for providing his report.

Reviewer 2 Report

The reviewed manuscript contains a brief overview of the application of probabilistic representation in quantum mechanics, to which the authors have made a significant contribution. This area of research is of great interest to the scientific community working on fundamental problems of quantum mechanics. Like any short review, it has some drawbacks, mainly related to the insufficiently detailed presentation and justification of the results. This remark applies mainly to the last sections, namely sections 6 to 8. In conclusion, it would also be desirable for the authors to outline the prospects more clearly for further research and applications.

In particular, in line 148, the variables X, μ and ν are replaced with variables with `tilde`, however, references to the necessary ratios are made after the transformed formula (86), which makes it difficult to understand due to the conciseness of the presentation.

Author Response

Following referee's recommendation, we added in the amended version to section 6 lines  133--140  and to section 8 lines 162--166.

Following referee's comment, in line 160 of the amended version we inserted formulas for \widetilde\mu and \widetilde\nu

Reviewer 3 Report

This article is very well written. I only have a few minor comments.

  1. Consistency: in paragraph 7, it says “In Sec. II, the tomographic probability approach is shortly reviewed……”; however, in the next paragraph it says” 2. Symplectic tomography“. The “2” should be changed to “II”; or the “II” in paragraph 7 should be changed to “2”.
  2. “V” in the Hamiltonian shown in eq. (5) should be potential energy. However, qE in line 61 is force. I noticed the right potential energy was given in a later part of this article. 
  3. English: “Hermitte” in line 88 should be “Hermite”.

Author Response

All corrections recommennded by the referee are done. We corrected numbering of sections in the Introduction and correct obvious misprints. After equation (5), according to referee's comment,  we remove a sentence with confusing interpretation of the potential energy.